# Insights into the Acid-Induced Gelation of Original Pectin from Potato Cell Walls by Gluconic Acid-*δ*-Lactone

**DOI:** 10.3390/foods12183427

**Published:** 2023-09-14

**Authors:** Dandan Lei, Likang Qin, Mei Wang, Haoxin Li, Zunguo Lei, Nan Dong, Jia Liu

**Affiliations:** 1School of Liquor and Food Engineering, Guizhou University, Guiyang 550025, China; dandanlei0202@163.com; 2Institute of Food Processing Technology, Guizhou Academy of Agricultural Sciences, Guiyang 550006, China; wm1375898692@163.com (M.W.); mcgrady456@163.com (J.L.); 3The Key Laboratory of Environmental Pollution Monitoring and Disease Control, Ministry of Education, School of Public Health, Guizhou Medical University, Guiyang 550025, China; haoxinli7523@163.com; 4Guizhou Key Laboratory of Agricultural Biotechnology, Guiyang 550006, China; dongnan1219@126.com

**Keywords:** potato pectin, gluconic acid-*δ*-lactone (GDL), gelation

## Abstract

The acid-induced gelation of pectin in potato cell walls has been gradually recognized to be related to the improvement in the cell wall integrity after heat processing. The aim of this study was to characterize the acid-induced gelation of original pectin from a potato cell wall (OPP). Rheological analyses showed a typical solution–sol–gel transition process of OPP with different additions of gluconic acid-*δ*-lactone (GDL). The gelation time (G_t_) of OPP was significantly shortened from 7424 s to 2286 s. The complex viscosity (η*) of OPP gradually increased after 4000 s when the pH was lower than 3.13 and increased from 0.15 to a range of 0.20~6.3 Pa·s at 9000 s. The increase in shear rate caused a decrease in η, indicating that OPP belongs to a typical non-Newtonian fluid. Furthermore, a decrease in ζ-potential (from −21.5 mV to −11.3 mV) and an increase in particle size distribution (from a nano to micro scale) was observed in OPP after gelation, as well as a more complex (fractal dimension increased from 1.78 to 1.86) and compact (cores observed by cryo-SEM became smaller and denser) structure. The crystallinity of OPP also increased from 8.61% to 26.44%~38.11% with the addition of GDL. The above results call for an investigation of the role of acid-induced OPP gelation on potato cell walls after heat processing.

## 1. Introduction

Potato pectin is a complex heteropolysaccharide, which mainly exists in the middle lamella between the primary cell wall of a potato [1] and plays a regulatory role in addressing the issue of the growth and cell shape of potatoes [2]. Extensive research has shown that potato pectin is a kind of low-methoxy (LM, DE < 50%) and highly acetylated pectin with a relatively low HG (homogalacturonan) structure and a high RG-I (type I rhamnogalacturonan) structure (>75%) [3,4,5]. HG is a galacturonic acid polymer connected by an *α*-1,4 glycosidic bond with esterified or aminated C-6 of galacturonic acid. RG-I is usually composed of repetitive units of rhamngalacturonic acid disaccharides with a variety of side chains, including arabinose and galactose [6,7]. In general, extraction methods could significantly affect the composition, structure, and rheological properties of pectin [8]. The chicory root pectin extracted by citric acid and sodium citrate in a study was a pseudoplastic fluid, while that extracted by ammonium oxalate and alkali was an expansive fluid [9]. Moreover, most studies have focused on the extraction of potato pectin with hydrolyzed chains by the aid of acidic, basic, and enzymatic hydrolysis for commercial applications [10,11,12,13]. These techniques have elevated the extraction efficiency of potato pectin by directly destroying the linkages among pectin, hemicellulose, and cellulose, accompanied with the destruction of the pectin structure. Recent research has revealed that original pectin in potato cell walls could participate in maintaining the cell structure during food processing [14,15,16]. It was found that the acid-induced in situ gelation of pectin in the cell wall played a role in the significant increase in the hardness of cooked potato slices. The pH or Ca^2+^ induced gelation of commercially extracted LM pectin from *Nicandra physalodes* (Linn.) Gaertn. seeds (NPGSP), *Premna microphylla turcz* (PMT), and *creeping fig* (CFP) has been extensively studied [17,18,19]. The gelation process of LM pectin in the presence of Ca^2+^ is often described as an “egg box” model [20,21,22]. Meanwhile, LM pectin can also form a gel through non-ionic binding based on hydrophobic interactions and hydrogen bond in the case of low pH [17,23].

So far, most studies have focused on the extraction and characterization of potato pectin with hydrolyzed chains for commercial purposes. However, little research has addressed the gelation process of original pectin from a potato cell wall (OPP). This study aimed at unraveling the gelation process of original pectin from a potato cell wall with the gradual variation in H^+^ (with the addition of d-glucono-*δ*-lactone). Thus, the cellulose and hemicellulose in the potato cell walls were hydrolyzed by cellulase to release original pectin. Through the addition of GDL, rheological features of solution–sol–gel transitions of original pectin were measured. Additionally, X-ray diffraction (XRD), dynamic light diffraction (DLS), small-angle X-ray scattering (SAXS), and low-temperature scanning electron microscopy (cryo-SEM) were used to explore the gelation process of original pectin from the potato cell walls.

## 2. Materials and Methods

### 2.1. Materials

The potatoes were purchased from Jinzhu Farmers Market (Huaxi District, Guiyang City, China). The *α*-amylase (food grade, 40,000 U/g) and mixed standards, including mannose (Man), ribose (Rib), rhamnose (Rha), glucuronic acid (GlcA), galacturonic acid (GalA), glucose (Glc), galactose (Gal), xylose (Xyl), arabinose (Ara), and fucose (Fuc), were purchased from Solarbio science & technology Co., Ltd. (Beijing, China), and the cellulase (BR, 400 U/mg) was obtained from Yuanye biological technology Co., Ltd. (Shanghai, China). Gluconic acid-*δ*-lactone (GDL, BR) was purchased from Macklin biological technology Co., Ltd. (Shanghai, China).

### 2.2. Methods 

#### 2.2.1. Preparation of Residue from Potato Cell Walls

Potatoes without pests and defects were selected for the experiment. After peeling, cleaning, and slicing, the potatoes were crushed with deionized water at a ratio of 1:1 (*w*/*v*, g/mL) for 1 min (32,000 r/min) by a breaker (Joyoung, China). The potato residue was repeatedly washed 3 times to remove starch and dried at 60 °C overnight for experiment. Then, deionized water was mixed with the potato residue at a ratio of 1:30 (*w*/*v*, g/mL). The mixture was adjusted to pH 6.6 ± 0.02, preheated at 95 °C for 5 min, treated with 5% (*w*/*w*, g/g) *α*-amylase (40,000 U/g), and kept in a water bath at 95 °C. After the mixture was tested for starch content with iodine solution and found to contain no starch, the residue was filtered and dried at 60 °C for 12 h. Then, ethanol (80%) was mixed with the residue at the ratio of 1:15 (*w*/*v*, g/mL). The mixture was heated at 75 °C for 25 min to remove any soluble sugar, filtered, and dried at 60 °C for the extraction of pectin.

#### 2.2.2. Enzymatic Extraction of Original Pectin from Potato Cell Walls

Original pectin from potato cell walls was extracted by enzyme, according to the method of Dranca et al. [24]. The residue was mixed with a citric acid–sodium citrate buffer (0.1 M, pH 4.8) at a ratio of 1:30 (*w*/*v*, g/mL) and preheated at 50 °C for 5 min. Then, 0.5% of cellulase (enzyme/residue, g/100 g) was added to the mixture and heated at 50 °C for 24 h. After the enzymatic hydrolysis of the residue, the mixture was boiled at 100 °C for 10 min. Upon cooling to room temperature, the filtrate was collected and adjusted to pH 6.5. Then, 80% ethanol was added to the filtrate at a ratio of 1:4 (*v*/*v*, mL/mL) and left overnight at 4 °C. The mixture was centrifuged at 7000 r/min for 10 min to acquire original pectin. After centrifugation, 80% ethanol was mixed with the residue (1:20, *w*/*v*) to resolve the soluble sugar. Finally, the mixture was centrifuged again at 7000 r/min for 10 min and freeze-dried. The yield of original potato pectin was calculated as
yield (%) = Pectin weight (g)/Weight of potato residue after starch removal (g) × 100%(1)

#### 2.2.3. Physicochemical Analysis of OPP

①Determination of molecular weight distribution (*M*_w_)

The molecular weight distributions of OPP were determined by size exclusion chromatography according to a previous method [17,18,19]. The separation of polymers was conducted on a Shimadzu LC-20A series HPLC system (Shimadzu Co., Kyoto, Japan) with a TSKgel G5000PWXL column (TOSOH, Tokyo, exclusion limits of 4000–800,000 Da, a particle size of 10 µm, and a pore size of 100 nm) and monitored by a reflective index detector (RI-20). The mobile phase composed of 0.1 mol/L NaNO_3_ and 0.06% NaN_3_ was used for the determination of *M*_w_ with a flow rate of 1.0 mL/min and column temperature of 40 °C, respectively.

②Determination of degree of methylation (DM) and acetylation (DA)

The DM of OPP was determined according to the method of Murayama et al. [25] with a little modification. OPP (20 mg) was first dissolved in 8 mL of distilled water and followed by an ultrasonic treatment (240 w, 10 min). The OPP solution was mixed with NaOH (2 mol/L, 3.2 mL) and vibrated at 25 °C for 1 h. Then, the OPP solution was mixed with HCl (2 mol/L, 3.2 mL) and vibrated at 25 °C for 15 min. A 0.1 mol/L PBS solution (KH_2_PO_4_-NaOH, pH 7.5) was added to the OPP solution to obtain a final volume of 25 mL. A 1 mL sample solution and different concentrations of methanol were mixed with 1 mL alcohol oxidase (1 U/mL) and incubated at 25 °C for 15 min. Then, a 2 mL pentanedione solution (25 g ammonium acetate, 3 mL acetic acid, and 0.25 mL acetylacetone) was added to the OPP solution and incubated at 58 °C for 15 min. After cooling to room temperature, the OPP solution was examined by an ultraviolet spectrophotometer at a wavelength of 412 nm. The DM of OPP was determined by using an acetic acid (RM) kit (Megazyme International Ireland Inc. Wicklow, Ireland) according to the product instruction.

③Determination of natural sugar ratio

The natural sugar ratio of OPP was determined by using HPLC (Shimadzu Co., Kyoto, Japan) with a Thermo BDS C18 column (250 × 4.6 mm i.d., 5 µm) according to a previous method [17,18,19]. OPP (2 mg) was hydrolyzed by 2 mL trifluoroacetic acid (2 M) at 110 °C in an oven for 3 h. Then, the sample was dried with N_2_ at 45 °C and redissolved in 200 μL of ultrapure water. The final pH of the sample was neutralized with 0.1 mol/L NaOH. OPP or a standard neutral sugar (400 μL), standard lactose (0.02 mol·L^−1^, 50 μL), NaOH (0.3 mol·L^−1^, 450 μL), and 1-phenyl-3-methyl-5-pyrazolone (0.5 mol·L^−1^, 450 μL) were mixed together by a vortex mixer (MX-S, DLAB Scientific Co., Ltd., Beijing, China). The mixture was incubated at 70 °C for 1 h, cooled to room temperature, and neutralized by the addition of HCl (0.3 mol·L^−1^, 450 μL). The sample was extracted by 1 mL chloroform 5 times, accompanied with vigorous mixing and centrifugation at 3000× *g* for 5 min each time. Then, the solution was filtered through a 0.45 mm membrane filter and followed by HPLC determination. Eluent A and B were composed of 15% acetonitrile with 0.05 M PBS solution (KH_2_PO_4_-NaOH, pH 7.1) and 40% acetonitrile with 0.05 mol/L PBS solution (KH_2_PO_4_-NaOH, pH 7.1), respectively. A gradient procedure (0–10 min, 0–10% B; 10–40 min, 10–40% B; 40–50 min, 40–0% B; 50–60 min, 0% B) was used for the separation of the sample. The signal was monitored with a wavelength of 245 nm, a flow rate of 0.7 mL·min^−1^, and a temperature of 35 °C. 

#### 2.2.4. Characterization Analysis of Acid-Induced Gelation Process and State of Original Pectin from Potato Cell Walls

A total of 0.3 g of pectin was dissolved in 10 mL distilled water and stirred at 60 °C for 40 min for full hydration. Different amounts of GDL (0.1%, 1%, 4%, 7%, and 10%, *w*/*v*) were added to the pectin solution (release rate of H^+^ was different) to control the gelation process. The pH variation of the pectin solution was recorded every 10 min. 

①Rheological measurements

For rheological property measurements, samples were immediately tested after the addition of GDL. The rheological characterization of samples was determined by using a DHR-2 rheometer (TA Instruments, New Castle, DE, USA), according to the method of Yuliarti et al. [26]. The conical plate (2° and 40 mm) was selected as the fixture. In order to eliminate the influence of mechanical action during sample loading on the test results, all samples were balanced and stabilized for 3 min before the test. Meanwhile, a strain range of 0.01–1000% was used to determine the linear viscoelastic region of all samples before the test. Amplitude scanning was performed at 1 Hz and 4 °C.

The dissolved samples were immediately tested on the rheometer for the observation of gelation time and measurement of dynamic frequency and static flow. Gelation time scan (150 min) was recorded at a temperature of 4 °C, frequency of 1 Hz, and strain of 1% (within the linear viscoelastic range of samples) for the pectin solution with different additions of GDL. The variation of energy storage modulus (G′), loss modulus (G″), complex modulus (G*), and complex viscosity (η*) during gelation time scan was used for the analysis of the gelation process. After the gelation time scan, the variation of G′ and G″ of the pectin solution was recorded at 4 °C and 1% strain with an oscillation frequency of 100 to 0.1 rad·s^−1^. After the frequency scan, the viscosity of the pectin solution was recorded at 4 °C and 1% strain with a shear rate ranging from 0.1 to 100 s^−1^.

②Dynamic light scattering (DLS) measurements

For other measurements, samples were tested after the addition of GDL for 2.5 h. The size characterization of samples was determined by using Zetasizer Nano ZS 90 (Malvern, UK) according to the method of Sun et al. [27]. Acid-induced pectin gelation was conducted without and with the addition of 4%, 7%, and 10% (*w*/*v*) GDL for 2.5 h. Then, the pectin solution (3%, *w*/*v*) was first pretreated with ultrasound at 240 W for 5 min before the test. The refractive index and temperature during the test were set at 1.33 and 25 °C, respectively.

③Small-Angle X-ray Scattering (SAXS) measurements

For other measurements, samples were tested after the addition of GDL for 2.5 h. The wet-state structural characterization of samples was determined by using Xeuss 2.0 Angle scatterer (Xenocs, Grenoble, France), according to the method of Alba et al. [28]. Acid-induced pectin gelation was conducted without and with the addition of 10% (*w*/*v*) GDL for 2.5 h. Then, the scanning of the sample (3%, *w*/*v*) was performed with a distance (between the sample and the detector) of 1007 mm, wavelength of 1.54189 Å, scattering vector (*q*) from 0.01 to 0.3 Å^−1^, optical tube power of 30 W, detector (Pilatus 3R) of 300 K, and size of a single pixel of 172 μm. During the measurement process, two scattering curves (sample scattering curve and solution background scattering curve) were obtained. The measured according to Equation (2), subtract the measured value from the background to eliminate interference. The final result of the SAXS was obtained as scattering intensity *I*(*q*) and scattering vector *q*.
*I*(*q*) = [*I*_S_, exp(*q*) − *I*_dc_(*q*)]/T_S_ − [*I*_M_, exp(*q*) − *I*_dc_(*q*)]/T_M_(2)

*q* stands for scattering vector, which is (4 *π*/*λ*) sin (*θ*); 2*θ* is the scattering angle; and I(*q*) is the scattering intensity of the corrected sample. *I*_S_, exp(*q*), *I*_dc_(*q*) and *I*_M_, and exp(*q*) are the sample measured scattering intensity, the detector current intensity, and the background measured scattering intensity, respectively. T_S_ and T_M_ are the X-ray transmittance of the sample and the solvent, respectively.

④X-ray Diffraction (XRD) measurements

For other measurements, samples were tested after the addition of GDL for 2.5 h. The dry-state structural characterization of the samples was determined by using D8 Advance X-ray diffractometer (Bruker, Germany), according to the method of Zhou et al. [29]. Acid-induced pectin gelation was conducted without and with the addition of 4%, 7%, and 10% (*w*/*v*) GDL for 2.5 h. Then, the pectin powder was acquired by freeze-drying. Scanning of the sample powder was performed using Cu Kα radiation with diffraction angles ranging from 5° to 90° (2*θ*), scanning speeds of 2°·min^−1^, voltage of 40 kV, and current of 40 mA.

#### 2.2.5. Cryo-Scanning Electron Microscopy (CSEM) Observations

For other measurements, samples were tested after the addition of GDL for 2.5 h. The morphological characterization of the samples was performed by using Regulus 8220 cryo-scanning electron microscope (Hitachi, Japan), according to the method of Kyomugasho et al. [30]. Acid-induced pectin gelation was conducted without and with the addition of 10% (*w*/*v*) GDL for 2.5 h. The pectin solution (3%, *w*/*v*) was first put into the sample table and quickly frozen for 30 s with liquid nitrogen slush. Then, it was transferred to the sample preparation chamber in a vacuum state for sublimation at −90 °C for 10 min and gold-plated at 10 mA for 60 s. Finally, the morphological observation of the samples was performed at −140 °C with an accelerating voltage of 5 kV.

### 2.3. Statistical Analysis

All measurement were repeated at least in triplicate. One-way analysis of variance (ANOVA) using Duncan’s multiple range test was used for evaluating the differences among groups with the SPSS statistics software (version 23.0). RheoCompass TRIOS software was used to obtain raw rheological data (including elastic modulus, viscous modulus, strain, and stress). In addition, fit 2D was used for transforming the SAXS result into a one-dimensional map.

## 3. Results and Discussion

### 3.1. Apparent Behavior of OPP without and with GDL-Induced Gelation

The yield of potato pectin (Table 1) obtained from the enzymatic method (cellulase hydrolysis) was slightly lower than that (14.34%) obtained from the chemical method (acid hydrolysis) [10]. The OPP showed a lower content (<50%) of GalA than pectin from other plants, indicating a lower content of HG, which is in accordance with previous studies [31,32]. Gal (68.674%) dominated the natural sugar content in OPP, suggesting that a branched structure existed in OPP [33,34]. The Rha/GalA of OPP (0.37) was significantly higher than that of citrus pectin (0.017~0.027), confirming that there were more branched chains on the OPP backbone [3,31]. Nevertheless, the (Ara+Gal)/Rha of OPP (17.4) was higher than that of commercial pectin (4.81), implying that a sophisticated branch structure existed in OPP [3,12]. OPP showed a higher Gal content, Rha/GalA, and (Ara+Gal)/Rha than those (49.38%, 0.11%, and 12.65%) of pectin extracted from potato by an acid (HCl) [3]. Moreover, OPP also showed a higher *M*_w_ (9.64 × 10^5^ g/mol) than that (2.80–3.20 × 10^5^ g/mol) of pectin extracted from potato by an acid (HCl, H_2_SO_4_, HNO_3_, citric acid, and acetic acid) [10]. The higher number of branched structures and higher *M*_w_ of OPP proved that pectin with natural characteristics was successfully extracted from potato cell walls.

GDL is one of the most widely used material for the acid-induced gelation of protein, such as soy protein isolate [35] and casein [36]. As shown in Figure 1, OPP showed varied flow morphologies from liquid to semisolid states with different addition amounts of GDL (0.1 to 10%). Thus, pH could significantly affect the gelation of OPP. Apparently, OPP with a 4% addition of GDL showed a transient state (sol) with enhanced adhesion from fluid to gel. With an excessive addition of GDL (more than 7%), OPP showed a gel-like state with poor mobility. The higher concentration made GDL release more H^+^ in the solution, which promoted the gelation of pectin [37]. Li et al. [20] also found that the addition of GDL (1.35%) into *Nicandra physalodes* (Linn.) Gaertn would cause the transformation of a gel from a weak to a strong state. Nevertheless, structural characteristics including molecular weight and esterification degree would significantly affect the gel mobility of pectin [38]. Compared with the acid-induced gelation of pectin (solid state with a hardness of 20–200 g) from other plants [20,39,40], OPP showed a weak gel with fluidity after gelation.

### 3.2. Rheological Analysis of OPP without and with GDL-Induced Gelation

Rheological analyses can simultaneously measure the characteristics of an elastic solid and a viscous fluid, which is suitable for studying the structure of complex fluids. However, a rheological analysis with a small deformation is not inhibited by structure in the measurement process and is often used to detect the transformation of hydrophilic food to a colloid sol–gel state [41,42]. GDL was used as a controlled-release H^+^ donor to gradually change the pH of the pectin solution. Thus, the gelation process of pectin could be easily captured through the analysis of the fluid characteristics. G′ and G″ represent the storage modulus (elastic part) and loss (viscous) modulus of fluid, respectively, which indicate the formation of structural or network bonding among polymer materials during the dynamic shear deformation process [41]. Typically, two fluid characteristics, denoted by G′ > G″ and G′ < G″, represent solid-like and liquid-like behavior, respectively. 

According to the time-scanned graph (Figure 2A–F), G′ increased significantly with the decrease in pH within 9000 s, inferring that OPP experienced a gradual formation of an enhanced structure with network bonding. The decrease in pH will lead to the reduction of the number of charged groups (carboxyl groups), thereby reducing the electrostatic repulsion between polymers [43] and promoting hydrogen bonding and hydrophobic interaction between pectin chains [19,40]. When the pH was higher than 4.19, G′ was consistently lower than G″, indicating a fluid state of OPP. However, OPP showed a typical gel formation process (G′ > G″) with the pH lower than 3.13. The definition of the sol–gel transition point is generally accepted at a given frequency. The cross-point between G′ and G″ was defined as gelation time (G_t_) [42] and shortened (from 7427 s to 2286 s) with the decrease in pH. A similar observation was made in the gelation of pectin from seeds of the *creeping fig* plant with the addition of GDL [39]. The G_t_ (406 s) of pectin from seeds of the *creeping fig* plant was less than that (2767 s) of OPP around pH 2.5 [39].

The complex viscosity η* = η′ − iη″ is composed of dynamic viscosity η′ = G″/ω and imaginary viscosity η″ = G′/ω, representing the flow resistance of the fluid. The greater η* is, the stronger the cohesion inside the fluid, causing poor liquidity [44]. As shown in Figure 2 (G), η* gradually increased with time when the pH was lower than 3.13, which indicated the variation of the solution–sol–gel state.

As shown in Figure 2H, the pH of the OPP solution with different additions of GDL (from 0.1 to 10%) gradually decreased with time and reached a final value of 6.07, 4.19, 3.13, 2.59, and 2.3 at 9000 s, which was consistent with previous results [37]. The partial hydrolysis of GDL to gluconic acid in the aqueous systems caused the decrease in pH. 

Figure 3 shows different dependence tendencies in the fluid characteristics of OPP and frequency. Both the G′ and G″ of OPP increased with the increase in angular frequency from 0.1 to 100 rad/s. A rapid increase in G′ and G″ could be observed in OPP with the pH higher than 3.13, while a relatively stable variation in G′ and G″ appeared in OPP with the pH lower than 2.59. Within the frequency range of 0.1–10 rad·s^−1^, G″ and G′ dominated the fluid characteristic of OPP when the pH was higher and lower than 3.13, respectively. Within the frequency range of 10–100 rad·s^−1^, the G′ and G″ of OPP nearly coincided when the pH was higher than 2.59. However, there was obvious space between the G′ and G″ of OPP when the pH was lower than 2.59. This suggests that OPP had a frequency dependence with the variation in pH. Nevertheless, the frequency dependence was weakened with the formation of a strong gel, indicating that the gel structure was becoming more stable [45]. Gilsenan et al. found that the G′ and G″ of pectin (with 31% esterification degree) gradually increased with the decrease in pH to 3, showing a typical dilute solution with strong frequency dependence [46]. When the pH was lower than 2, the frequency scan result represented a typical strong gel with weak frequency dependence. Figure 3G shows that η of the gelled OPP significantly decreased (pH lower than 3.13) with the increase in the shear rate, while that of the non-gelled OPP (pH higher than 3.13) first increased in the low shear rate region from approximately 0.05 to 0.1 Pa·s and then decreased. The decrease in η with the increase in shear rate confirmed that OPP belongs to a typical non-Newtonian fluid (shear thinning). This may be ascribed to the formation of new conformation through the rearrangement and dissociation of polysaccharides along the flow direction [28]. Moreover, the highly branched structure of OPP might also contribute to the formation of chain entanglement, causing higher pseudoplastic characteristics [6,47].

### 3.3. DLS Analysis of OPP without and with GDL-Induced Gelation

Generally, the particle characteristics (ζ-potential and particle size distribution) represent the stability and aggregation behaviors of pectin polysaccharide [48]. The ζ-potential of pectin represents the potential stability of the colloidal system. As can be seen from Figure 4A, negative ζ-potential values were observed in OPP before and after gelation. Carboxylate ions (-COOH) on the skeletal structure mainly contributed the surface negative charge of pectin [43,49]. With the decrease in pH, the ζ-potential of OPP decreased from −21.5 mV to −11.3 mV, which was also observed in the acid-induced gelation of citrus pectin [50] and *Nicandra physalodes* (Linn.) pectin [51]. Smaller particles usually share a more specific surface area, providing a higher ζ-potential absolute value for polymers [52]. The decrease in ζ-potential in OPP after acid-induced gelation can be ascribed to the surface protonation that occurred with the carboxyl group at a lower pH. The reduction in the surface electric charge would trigger the aggregation of pectin through a reduction in the electrostatic repulsion [44,53]. Figure 4B shows that the OPP particles mainly dispersed in water around 200 nm. With the decrease in pH, the particles of OPP experienced a transformation from a narrow to a broad distribution, and from a nano to micro scale (Figure 4B–E). The elevated distribution and size of OPP in solution might be due to the aggregation of polymers, which is caused by the decrease in ζ-potential [54]. Nevertheless, the increase in particle size after gelation might further improve the viscosity and decrease the colloidal fluidity (observation in Figure 1) of pectin through an elevated hydration ability [55].

### 3.4. SAXS and XRD Analysis of OPP without and with GDL-Induced Gelation

SAXS is an accurate and a non-destructive analytical method for characterizing the structural features of polymers [56]. Figure 5 shows the two-dimensional scattering diagram of OPP and gelated OPP (10% GDL-induced gelation). As shown in Figure 5A, a circular scattering pattern was observed in OPP. In contrast, more scattering highlights appeared in the GDL-induced gelation of OPP, creating a typical spectrogram of a hydrogel with weak anisotropy [57]. The fractal dimension can quantitatively describe the binding state of the system and the complexity of the polymer network [58]. The fractal dimensions of OPP (1.78) and gelated OPP (1.86) were obtained by a linear fitting of the linear segment of the one-dimensional curve. A loose structure of a polymer corresponds to a smaller fractal dimension, while a tight structure corresponds to a larger one [59]. An acidic environment was adverse to the formation of a tighter chain arrangement in pectin, especially for that containing an amount of neutral sugar side chains (RG-I regions). X-ray diffraction is a powerful tool used to identify the crystal structure of polymers [60]. Normally, natural pectin contains heteropolysaccharides, causing a semi-crystal or amorphous structure [61]. Figure 5 shows a wide diffraction peak near 2*θ* of 20.71° in OPP powder. Interestingly, the intensity of the 20.71° diffraction peak significantly increased after the gelation of OPP. This result is in accordance with the GDL-induced gelation of *Nicandra physalodes* (Linn.) Gaertn. seeds (NPGSP) pectin, which also shows an intense peak near 2*θ* of 20.97° [20]. Nevertheless, the crystallinity of OPP increased from 8.61% to 26.44%~38.11% with the addition of GDL. The increase in pectin crystallinity after gelation was probably due to the formation of ordered structures during the molecular interactions, such as hydrophobic effect and hydrogen bond [44,62].

### 3.5. In Situ Observation of OPP without and with GDL-Induced Gelation

Traditional SEM requires freeze-drying the sample in advance, which easily causes damage to the morphology during the process of ice crystal freezing/thawing [63]. However, cryo-SEM could easily obtain the original morphology for the analysis of colloids due to the reduction of deformation in pretreatment [30]. Figure 6 shows the cryo-electron microscope scanning images of OPP and gelated OPP (10% GDL-induced gelation). OPP showed a porous structure with many large holes (>5 μm). In contrast, a compact structure was observed in gelated OPP with many small holes (<5 μm). Abundant arabinose side chains produced more entanglement forces among side chains in OPP through hydrophobic interactions and hydrogen bonds, which also limited and promoted the mobility of pectin and the formation of a stable gel network, respectively [64,65].

## 4. Conclusions

The characteristics during and after acid-induced gelation of OPP by the addition of GDL was investigated. The gelation of OPP was first observed at a pH lower than 3.13. A further decrease in pH to 2.59 caused the formation of gel-like substances and increased the complex viscosity (η*) from 0.15 to a range of 0.20~6.3 Pa·s at 9000 s. The gelation time (Gt) of OPP was significantly shortened from 7424 s to 2286 s. The complex viscosity (η*) of OPP gradually increased after 4000 s when the pH was lower than 3.13. Moreover, the gelation of OPP caused the decrease in ζ-potential (from −21.5 mV to −11.3 mV), which promoted the protonation of carboxyl groups on the surface of OPP and reduced the electrostatic repulsion among OPPs. Thus, the aggregation of pectin with a wider particle size distribution (from a nano to a micro scale) and mutual linked structures (the fractal dimension of OPP increased from 1.78 to 1.86, and the crystallinity of OPP increased from 8.61% to 26.44%~38.11%) was observed after gelation of OPP, which might contribute to the enhanced mechanical properties of potato cell walls during the heating process.

## Figures and Tables

**Figure 1 foods-12-03427-f001:**
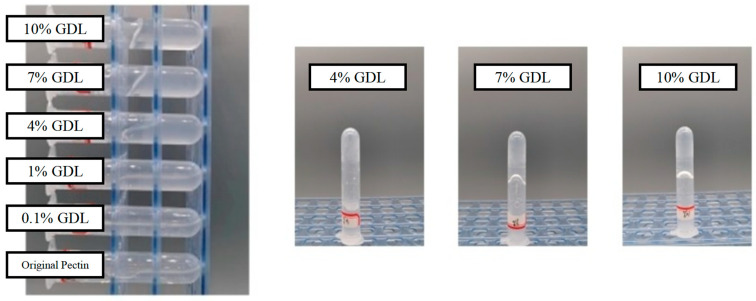
Appearance of GDL (0.1–10%) induced gelation of OPP.

**Figure 2 foods-12-03427-f002:**
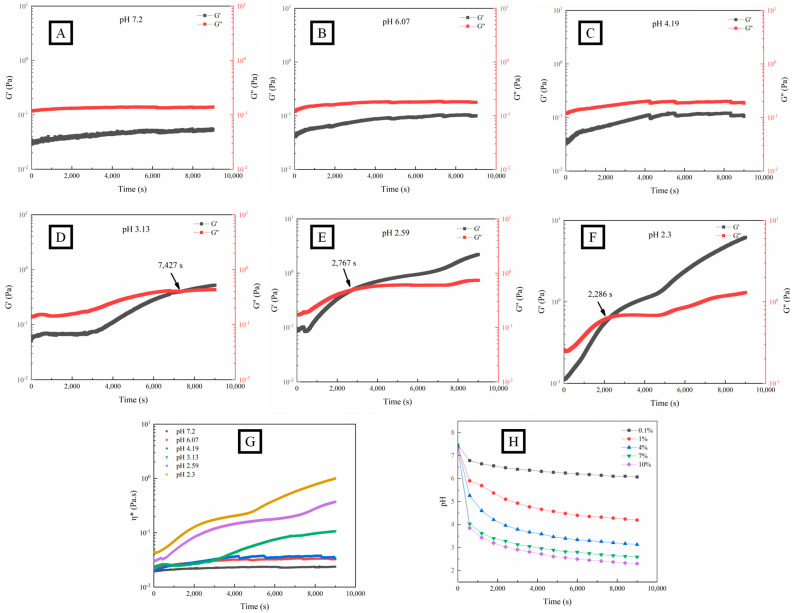
Fluid characteristics (G′, G″, and η*) and pH variation (**H**) of OPP without (**A**) and with the addition of GDL ((**B**–**G**), 0.1–10%) at fixed strain (1%) and frequency (1 Hz).

**Figure 3 foods-12-03427-f003:**
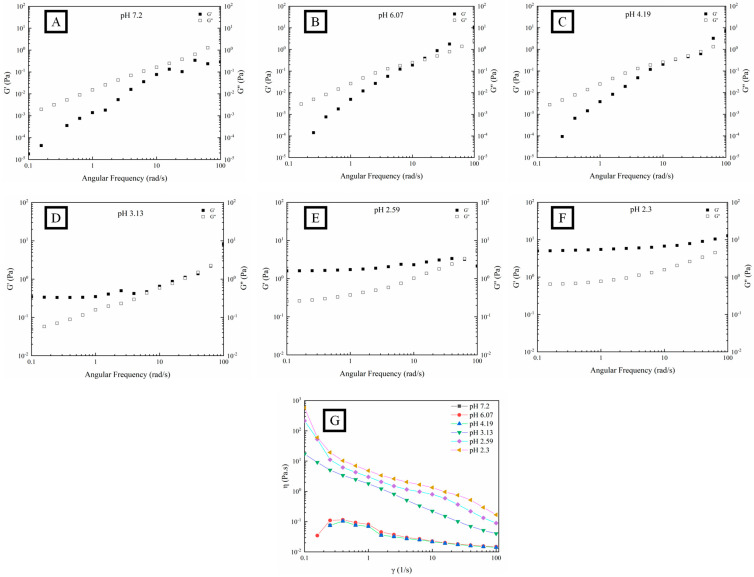
Fluid characteristics (G′, G″, (**A**–**F**) and η, (**G**)) of OPP without (**A**) and with the addition of GDL ((**B**–**F**), 0.1%–10%) at fixed strain (1%).

**Figure 4 foods-12-03427-f004:**
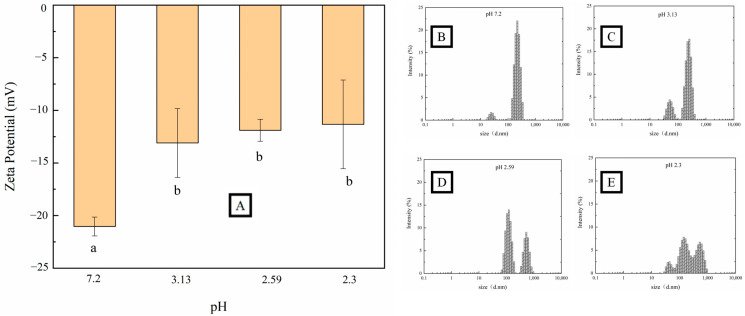
Particle characteristics (ζ potential, (**A**) and particle size distribution, (**B**–**E**)) of OPP without (**B**) and with the addition of GDL ((**C**–**E**), 4%–10%). (a and b represents a significant difference between adjacent variables, *p* < 0.05).

**Figure 5 foods-12-03427-f005:**
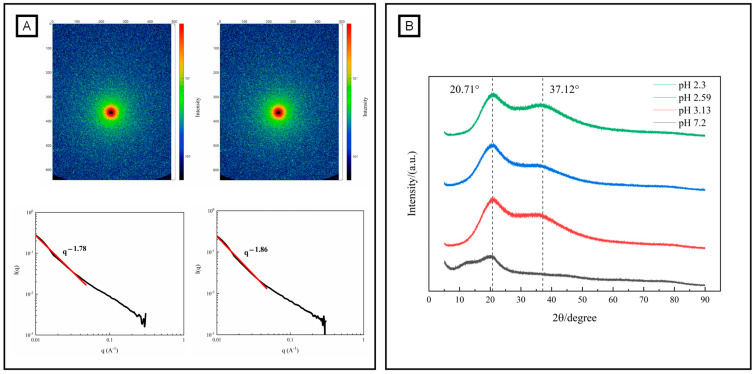
Structural characteristics (fractal, (**A**) and crystal, (**B**) structure) of OPP without and with the addition of GDL. (black and red line in (**A**) represents one-dimensional curve and linear fitting of the linear segment of the one-dimensional curve).

**Figure 6 foods-12-03427-f006:**
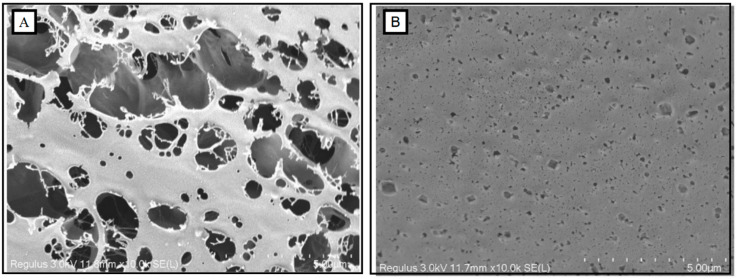
Morphological characteristics (cryo-SEM, with 10,000× magnification) of OPP without (**A**) and with the addition of GDL ((**B**), 10%).

**Table 1 foods-12-03427-t001:** Structural characterization of potato pectin.

Composition	Content
Yield (dry basis, %)	13.03 ± 0.03
Degree of methylation (DM, %)	24.74 ± 0.01
Degree of acetylation (DA, %)	7.75 ± 0.20
Molecular weight (*M*_w_, g/mol)	9.64 × 10^5^
Mannose (Man, %)	0.35 ± 0.12
Ribose (Rib, %)	0.06 ± 0.01
Rhamnose (Rha, %)	4.48 ± 0.38
Glucuronic acid (GlcA, %)	0.44 ± 0.11
Galacturonic acid (GalA, %)	12.19 ± 0.27
Glucose (Glc, %)	3.71 ± 0.24
Galactose (Gal, %)	68.66 ± 0.14
Xylose (Xyl, %)	0.09 ± 0.01
Arabinose (Ara, %)	9.67 ± 0.25
Fucose (Fuc, %)	0.34 ± 0.06
Rha/GalA	0.37
(Ara+Gal)/Rha	17.48

## Data Availability

The data presented in this study are available on request from the corresponding author.

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
