# Peer review of "Insights into the Acid-Induced Gelation of Original Pectin from Potato Cell Walls by Gluconic Acid-δ-Lactone"

_foods, 2023, doi:10.3390/foods12183427_

Round 1
Reviewer 1 Report
Please check the attached file; reviewer comments-Foods

English language requires minor improvements as some sentences are incomplete and should be clear.
Reviewer 2 Report
Dear authors
Thank you for your interesting and very carefully written work I found only a few errors in it - in writing and in the preparation of drawings
Please correct this
typo errors: Line 79/80 – interrupted sentence on line 79
Page 5, fig 2 - to keep the logic of the sequence, change the order of the figures in fig. 2 - pH 7 / pH 6 / pH 4 / pH 3 / pH 2.59 / pH 2.3. currently pH 2.59 is ahead of pH 3 - which introduces difficulties in observing phenomena and in observing the shift of the float point from pH. Figure 3 has already been done in the correct (logical) order)
Fig 5 - please label the lines in the drawing exactly. The current signature uses thin, colored lines whose color simply cannot be seen. Also, in the caption pH 7 is at the top - while these experimental data are at the bottom of the figure. And exactly the same reversal of the order of the description and the order of the data presented occurs for all pH
Reviewer 3 Report
the authors extracted pectin from potato and used further to investigate gelation properties. The results presented in the manuscript are interesting, and I reccommend some minor revision:
a) Please, include the table A1 (apendix) in main text, and in experimental part write the detailled procedure of characterization of extracted pectin.
b) Compare the yield, DM, etc-other properties of extracted pectin with the data from literature. What is advantage and disadvantage of etraction process used in this work, and in pectin properties obtained in this work, in coresspondance with the literature.
c) Discuss gelling properties of pectin obtained in this work with gelling properties of other pectins, and with other acids, too.
Minor revision is required.
